# Long-Term Outcome of Critically Ill Advanced Cancer Patients Managed in an Intermediate Care Unit

**DOI:** 10.3390/jcm11123472

**Published:** 2022-06-16

**Authors:** Nerea Fernández Ros, Félix Alegre, Javier Rodríguez Rodriguez, Manuel F. Landecho, Patricia Sunsundegui, Alfonso Gúrpide, Ramón Lecumberri, Eva Sanz, Nicolás García, Jorge Quiroga, Juan Felipe Lucena

**Affiliations:** 1Department of Internal Medicine, Division of Intermediate Care and Hospitalists Unit, Clinica Universidad de Navarra, 31008 Pamplona, Spain; falegre@unav.es (F.A.); mflandecho@unav.es (M.F.L.); psunsundegu@unav.es (P.S.); ngarcia@unav.es (N.G.); jquiroga@unav.es (J.Q.); flucena@unav.es (J.F.L.); 2Department of Oncology, Clinica Universidad de Navarra, 31008 Pamplona, Spain; jrodriguez@unav.es (J.R.R.); agurpide@unav.es (A.G.); 3Navarra Institute for Health Research (IdiSNA), 31008 Pamplona, Spain; 4Hematology Service, Clinica Universidad de Navarra, 31008 Pamplona, Spain; rlecumber@unav.es; 5Faculty of Medicine, European University of Madrid, 28670 Madrid, Spain; esanz@centrogf.com; 6Centro de Investigación Biomédica en Red Enfermedades Hepáticas y Digestivas (CIBERehd), 28801 Madrid, Spain

**Keywords:** advanced cancer, intermediate care unit, do not resuscitate orders, acute complications, hospitalists

## Abstract

Background: To analyze the long-term outcomes for advanced cancer patients admitted to an intermediate care unit (ImCU), an analysis of a do not resuscitate orders (DNR) subgroup was made. Methods: A retrospective observational study was conducted from 2006 to January 2019 in a single academic medical center of cancer patients with stage IV disease who suffered acute severe complications. The Simplified Acute Physiology Score 3 (SAPS 3) was used as a prognostic and severity score. In-hospital mortality, 30-day mortality and survival after hospital discharge were calculated. Results: Two hundred and forty patients with stage IV cancer who attended at an ImCU were included. In total, 47.5% of the cohort had DNR orders. The two most frequent reasons for admission were sepsis (32.1%) and acute respiratory failure (excluding sepsis) (38.7%). Mortality in the ImCU was 10.8%. The mean predicted in-hospital mortality according to SAPS 3 was 51.9%. The observed in-hospital mortality was 37.5% (standard mortality ratio of 0.72). Patients discharged from hospital had a median survival of 81 (30.75–391.25) days (patients with DNR orders 46 days (19.5–92.25), patients without DNR orders 162 days (39.5–632)). The observed mortality was higher in patients with DNR orders: 52.6% vs. 23.8%, *p* 0 < 0.001. By multivariate logistic regression, a worse ECOG performance status (3–4 vs. 0–2), a higher SAPS 3 Score and DNR orders were associated with a higher in-hospital mortality. By multivariate analysis, non-invasive mechanical ventilation, higher bilirubin levels and DNR orders were significantly associated with 30-day mortality. Conclusion: For patients with advanced cancer disease, even those with DNR orders, who suffer from acute complications or require continuous monitoring, an ImCU-centered multidisciplinary management shows encouraging results in terms of observed-to-expected mortality ratios.

## 1. Introduction

In recent years, new diagnostic tools and an increasing number of anticancer drugs have revolutionized the outcomes for oncologic patients, leading to a great improvement in survival [1]. These improvements run in parallel with an increasingly complex medical management due to adverse events derived from cancer or from its treatment. In fact, up to 15–20% of admissions in the intensive care unit (ICU) are currently patients with cancer [2,3,4]. In this population, where a non-defective percentage have an advanced disease and/or *do not resuscitate orders* (DNR orders), achieving the balance between avoiding overtreatment, a too early treatment withdrawal and cost-effectiveness become key elements for good clinical practice.

In this context, intermediate care units (ImCUs) may offer a suitable alternative to ICUs for those advanced cancer patients with acute complications whose admittance to the ICU is precluded (e.g., patients with *DNR orders* or contraindications for mechanical ventilation and/or renal replacement therapy) but whose required care surpasses that provided in a general ward. Previous data have shown that ImCUs can reduce costs, improve ICU utilization for sicker patients, decrease ICU readmissions, promote greater flexibility in patient triage, and decrease mortality rates in hospital wards [5,6,7,8,9]. However, data from patients with advanced cancer (stage IV) admitted to an ImCU are so far lacking in the literature. The objective of this study was to analyze the main patient characteristics, clinical outcomes and survival times of advanced cancer patients admitted to an ImCU led by hospitalists in a third level academic hospital. Since DNR orders may substantially modify the therapeutic approach, a specific analysis of this subgroup of patients was carried out.

## 2. Material and Methods

We performed a retrospective observational study with data collected from April 2006 to January 2019 in a single academic medical center in Pamplona, Spain. All patients included in the study had advanced (stage IV) cancer (solid or hematological) and were admitted to our ImCU under different medical or surgical circumstances. Patients with a follow-up of less than 30 days after hospital discharge were excluded from the analysis.

In our institution, the ImCU is a 9-bed unit adjacent to, but independent from, the mixed ICU. Each bed is fully equipped with continuous telemetry, pulse oximetry, non-invasive arterial blood pressure, central venous pressure monitoring and non-invasive pressure support ventilation. The signals are relayed to a central monitoring station. The ImCU infrastructure (beds, technical resources and nursing staff) is shared with the stroke unit and the coronary care unit. The nurse–patient ratio is 1:3. The ImCU rounding team involves a nurse, the hospital pharmacist, the ImCU resident, the specialist or surgeon and the attending hospitalist. Hospital medicine is a medical specialty dedicated to the delivery of comprehensive medical care to hospitalized patients. Thus, a hospitalist is a physician whose primary professional focus is the general medical care of hospitalized patients.

The hospitalist is responsible for admission and discharge of all ImCU patients. Admission and discharge criteria were set according to previous guidelines defined by The American College of Critical Care Medicine [10]. Exclusion criteria from ImCU admission were: age less than 18 years, severe respiratory failure at imminent risk of intubation, status epilepticus and catastrophic brain illness. The hospitalist usually ordered diagnostic or therapeutic interventions as needed, with the exception of orders for procedures or consultations related with specialist’s specific needs. All the patients admitted to the ImCU were co-managed with the respective physician in charge (medical oncologist, general surgeon, hematologist, etc.).

Demographics, past medical history, type of cancer and prior therapies, reason for admission to ImCU, physiological parameters and laboratory variables (at the time of admission), as well as length of stay (LOS), length of stay in ImCU (LOS ImCU), functional status (independent, partially dependent or totally dependent for activities of daily living), ECOG performance status, Charlson comorbidity index score of each patient and number of patients with do not resuscitate orders were recorded. Patients with DNR orders are an especially frail population, and thus we also aimed to know their outcomes after their admittance to the intermediate care unit.

The Simplified Acute Physiology Score 3 (SAPS 3) was used as a prognostic and severity score [11,12,13]. This score was developed in 2005 in a worldwide prospective study to predict in-hospital mortality in patients admitted to ICUs [11,12]. Soares et al. [14] conducted the first external validation of the SAPS 3 in critically ill cancer patients, showing a high accuracy in the prediction of in-hospital mortality. This score has also been validated in patients with cancer by several groups [15,16].

In-hospital mortality, 30-day mortality and survival after hospital discharge (in days) for each patient were calculated (31 December 2019 was set as the final date of follow-up). In order to calculate survival from hospital discharge in those patients whose date of death was unknown, last visit date was set as a reference.

SPSS for Windows, version 20.0 (SPSS Inc., Chicago, IL, USA) was used for statistical analysis. All data are presented as frequencies (percentage) for qualitative variables and means (standard deviation) for quantitative variables. Univariate analysis was performed using t-Student or chi-square test, as appropriate. The SAPS 3 scores with its respective predicted mortality rates were calculated according to standard coefficients [12,17]. Logistic regression was performed to identify independent prognostic variables for in-hospital mortality and 30 days after discharge mortality. Variables included in the multivariable model were determined by using univariable logistic regression (*p*-value in the univariable model <0.20).

The institutional review board at the Clínica Universidad de Navarra in Pamplona, Spain approved the study protocol. The study was performed in accordance with the ethical principles of the 1964 Declaration of Helsinki and its later amendments.

## 3. Results

Two hundred and forty patients with advanced (stage IV) cancer were included in the study. The mean length of stay at the ImCU was 23.7 days. One hundred and fifty-nine patients (66.3%) were male and the mean age was 62.5 years old. Most patients (83.8%) had a Charlson comorbidity index score >7 and 77.1% of patients had an ECOG performance status greater than or equal to 2. One hundred and thirty three patients (56.2%) were partially dependent and a third of patients were dependent for basic activities of daily life. Gastrointestinal (gastrointestinal plus hepatobiliary) and lung cancer were the most predominant (33.8% and 20.8%, respectively) diagnoses. One hundred and fourteen patients (47.5%) of the cohort had do not resuscitate orders. Table 1 summarizes the main characteristics of the study population.

Admission to the ImCU was urgent in 87.5% of patients. In total, 59.6% of cases came from the conventional ward and 30.8% came from the emergency department. Up to 7.5% of the patients were transferred from the ICU to the ImCU. The two most frequent reasons for admission to the ImCU were sepsis (77 patients, 32.1%) and acute respiratory failure (excluding sepsis) (93 patients, 38.7%). One hundred and fifty-seven patients (65.4%) had an SpO2 <90% regardless of the main reason for admittance to the ImCU. Non-invasive mechanical ventilation and the use of vasopressors were required in 36.7% and 10.8% of the patients, respectively.

Mortality in the ImCU was 10.8%. The mean predicted in-hospital mortality according to SAPS 3 was 51.9%. However, the observed in-hospital mortality in our patients was 37.5% (90/240) resulting in a standardized mortality ratio of 0.72. The need for an urgent admission to the ImCU was not associated with a higher ImCU mortality (11.4% vs. 6.7%, *p* = 0.432), in-hospital mortality (37.6% vs. 36.7%, *p* = 0.92), or 30-day mortality (52.4% vs. 50%, *p* = 0.80) compared to those observed in patients without an urgent admission.

Seventy percent of the patients admitted to the ImCU achieved a significant clinical improvement that allowed discharge to the general ward after a mean length of stay of 5.1 days in the ImCU. Twenty-one patients (8.7%) were transferred to the ICU (17 due to a worsening of their general condition and four because surgery was indicated). Six of these twenty-one patients were discharged from hospital after recovery of their acute complication. Overall, 150 patients were discharged from hospital with a median survival of 81 (30.75–391.25) days. In total, 48%, 34.7% and 25.3% of the patients were alive at 3, 6 and 12 months after discharge, respectively (Table 2).

### 3.1. DNR vs. No-DNR Orders

Patients with DNR orders were significantly older, had a higher Charlson comorbidity index score, a worse ECOG performance status and were slightly more dependent. There were no differences in the percentage of patients with any degree of respiratory insufficiency at admission and the overall length of stay among patients with and without DNR orders. When we analyzed the two main causes (*sepsis and acute respiratory failure) compared to other causes,* there were no differences between both groups. On the other hand, patients with DNR orders had a longer length of stay in the ImCU (5.8 vs. 4.5 days, *p* = 0.023) and a more frequent use of both non-invasive mechanical ventilation and vasoactive drugs (Table 3).

The SAPS 3 score and, thus, the predicted in-hospital mortality were significantly higher in those patients with DNR orders compared to those without them: 59.8% vs. 44.8%, respectively. The observed mortality was also significantly higher in patients with DNR orders: 52.6% vs. 23.8%, *p* <0.001. In both cases, the standard mortality ratio was less than 1 (0.87 vs. 0.53). Mortality in the ImCU was also significantly higher in patients with DNR orders (21.1% vs. 1.6%, *p* < 0.001). Fourteen percent of patients with DNR orders survived more than 3 months after discharge and almost 6% were alive after 6 months. The median overall survival was significantly longer in those patients without DNR orders than those without DNR orders: 162 (39.5–632) vs. 46 (19.5–92.25) days, *p* < 0.001. (Table 3).

The subgroup of patients with DNR orders, ECOG 0–2 and being on immunosuppressive treatment (*steroids, chemotherapy or other immunosuppressive treatment six months prior to admission*) were independently associated with being alive at 3 months after discharge (data not shown).

### 3.2. Prognosis Factors for In-Hospital Mortality

Univariate analysis showed an increase in in-hospital mortality for those patients with cirrhosis, DNR orders, worse ECOG performance status, need for non-invasive mechanical ventilation, those on any immunosuppressive treatment (*steroids, chemotherapy or other immunosuppressive treatment—e.g., rituximab, methotrexate*), lower platelet count, higher urea and bilirubin level and higher SAPS 3 Score) (see Appendix A). Those who did not die during hospitalization had a mean survival of 289 days.

Multivariate logistic regression showed an increased in-hospital mortality for those with higher ECOG performance status (3–4 vs. 0–2), higher SAPS 3 score, and DNR orders (Table 4). We excluded those variables included in the SAPS 3 score to avoid collinearity.

In patients with DNR orders (*n* = 114), ECOG performance status (3–4 vs. 0–2) (*p* = 0.034) and the presence of COPD (*p* = 0.038) were the only factors significantly associated with a higher risk of in-hospital death by multivariate logistic regression.

### 3.3. Prognosis Factors for 30-Day Mortality

The SAPS 3 score was excluded from the analysis given that this score was developed for in-hospital mortality. The univariate analysis showed a significantly higher mortality for those patients with cirrhosis, DNR orders, worse ECOG performance status, need for non-invasive mechanical ventilation, longer length of stay in the ImCU and higher leucocytes, bilirubin and urea levels (see Appendix A). Those who did not die during this period had a mean survival of 373 days.

Multivariate analysis showed an increase in 30-day mortality for those who required non-invasive mechanical ventilation, with higher bilirubin levels and DNR orders (Table 5).

## 4. Discussion

The outcomes for advanced cancer patients after ICU admission, in terms of in-hospital mortality, remain controversial. While some authors suggest that metastatic disease does not interfere with in-hospital mortality after ICU admittance [18,19], other groups have reported advanced disease as an adverse prognostic factor for in-hospital mortality [19] or as a short-term mortality risk factor [20]. Therefore, the admittance to the ICU of critically ill patients with advanced cancer is currently a matter of intense debate [21,22]. ImCUs have emerged as a potential effective alternative to ICUs in this setting. To the best of our knowledge, this is the first study describing the outcomes and prognostic factors for advanced cancer patients admitted to an ImCU led by hospitalists.

We used the Simplified Acute Physiology Score 3 (SAPS 3) as a prognostic model score. The mean SAPS3 score observed in our population was 68 points (predicted in-hospital mortality of 51.9%). This figure exceeds those reported for cancer patients admitted to ICUs [14,15,16,23,24], suggesting that our population is a valid representation for this group of patients for the purpose of the study. In fact, several features reinforced the frailty and illness severity of our patient population: non-invasive mechanical ventilation and vasoactive drugs requirements in 36.7 and 10.8% of the patients, respectively, a Charlson comorbidity index score ≥ 7 in 83.8% of the patients, a partial or complete dependency in 56.2 and 31.7%, respectively, and an ECOG ≥ 2 in 77% of them. Despite this high rate of adverse prognostic factors, the ImCU mortality was 10.8%. This percentage is significantly lower than that reported by other studies with similar patient cohorts admitted to ICUs [2,15,16,19,23,24,25,26,27]. It is noteworthy that the studied population was relatively young (mean age 62.5 years old). It could be thought that this may have influenced the observed mortality rate. However, despite the age, the frailty of this population addressed by ECOG performance status and the Charlson comorbidity index score was shown. Moreover, the SAPS 3 score includes age and, as we said above, the SAPS 3 score in our population was higher than in other studies conducted with cancer patients in critical care units. In our population, age was not associated with 30-day mortality, highlighting the idea that other variables such as ECOG performance status may have more impact in mortality than age *per se*. Although it is out of the scope of the study and we did not analyze it specifically, the potential impact of a multidisciplinary approach based on the co-management of all these complex patients between the hospitalist and the specialist in charge warrants further research.

With a mean length of stay of 5 days in the ImCU, 70% of patients were discharged due to an improvement in their status and/or resolution of the acute complication. Up to 62.5% of the patients were discharged from hospital with a standardized mortality ratio of 0.72. This ratio is similar to other results in cancer patients that included *all stages* [14,15,16,23,24,26] and to the overall non-cancer population [11,12] admitted to ICUs without admittance restrictions. The median survival was 81 days (30.7–391.2) and almost 25% of the cohort was alive more than 6 months after discharge. In-hospital mortality did not differ according to the cause of admission to the ImCU or the urgency of the admission.

As expected, a higher SAPS 3 score was an independent risk factor for in-hospital mortality, among others. Need for non-invasive mechanical ventilation was an independent risk factor for 30 days after discharge mortality. DNR orders were associated with both in-hospital and 30 days after discharge mortality. The 114 patients with DNR orders in our cohort were older, with a higher Charlson comorbidity index score and a worse ECOG performance status, and were more dependent compared to those patients without DNR orders. Their SAPS 3 score and in-hospital mortality were also significantly higher, although the standardized mortality ratio was still less than 1 (0.87). Overall observed mortality was similar to other reports in advanced cancer patients [26] or in those not eligible for intensive care units due to age or multiple comorbidities [28]. Taking into account the frailty of these patients and the poor prognosis according to their advanced cancer disease, in our opinion, the present results reinforce the utility of the ImCU in the management of acute complications within this highly complex population, achieving a good balance between futile therapeutic maneuvers and nihilism, and providing a cost-effective alternative to ICUs. Even more, the impressive progress of anticancer therapies during the last decade has allowed prognostic improvement for patients compared with some years ago when they would not have been eligible for conventional anticancer drugs due to their performance status and frailty. In this setting, the ImCU may also become a suitable alternative to treat acute complications derived from increasingly used novel anticancer therapies such as immune checkpoint inhibition, T-cell engaging therapies such as bispecific T-cell engaging (BiTE) single-chain antibody constructs and chimeric antigen receptor (CAR) T cells, bispecific antibodies or antibody–drug conjugates. These complications, mostly related to a cytokine release syndrome, may be life threatening in 10–15% of the patients and their management requires prompt and highly close monitoring unlikely to be provided in a conventional general ward [29].

Regardless of “*DNR status*”, it is noteworthy that 7.5% of patients admitted to the ImCU were downstaged from the ICU. This highlights also the suitability of the ImCU for those patients whose clinical needs exceed what the general ward can offer.

Our study has several limitations: First, its retrospective nature; second, we do not know whether the outcomes of these patients would have been different if they were managed in conventional wards instead of the ImCU. However, it seems reasonable to think, based on published data, that those patients managed in the ImCU would have a better short-term survival [30,31]. Third, there may be a certain grade of overestimation in mortality in some cases due to the heterogenous case mix (type of cancer). However, the proportion of patients with more favorable types of cancer (e.g., neuroendocrine tumor) was small.

In conclusion, our data show that selected patients with advanced cancer disease, even those with DNR orders, who suffer from acute complications or require continuous monitoring, may be co-managed properly in ImCUs, showing encouraging results in terms of observed-to-expected mortality ratios.

## Figures and Tables

**Table 1 jcm-11-03472-t001:** General patient characteristics.

Variable	
Patients, *n*	240
Male, *n* (%)	159 (66.3)
Age	62.5 (13.2)
BMI	24.6 (4.7)
Diabetes Mellitus, *n* (%)	53 (22.1)
CHF, *n* (%)	21 (8.8)
Ischemic heart disease, *n* (%)	17 (7.1)
COPD, *n* (%)	40 (16.7)
Hypertension, *n* (%)	85 (35.4)
Cirrhosis, *n* (%)	18 (7.5)
Charlson comorbidity index score, *n* (%)	
● 0–4 4–6 >7	9 (3.7) 30 (12.5) 201 (83.8)
ECOG, *n* (%)	
● 0–1 2 3 4	55 (22.9) 107 (44.6) 65 (27.1) 13 (5.4)
Functional status, *n* (%)	
● Independent Partially dependent Dependent	29 (12.1) 135 (56.2) 76 (31.7)
Type of cancer, *n* (%)	
● Solid tumor ○ Gastrointestinal HPB Gynecological Breast Genitourinary/Renal Lung Neurological ENT Neuroendocrine Other Hematological malignancy	229 (95.4) 46 (19.2) 35 (14.6) 12 (5) 14 (5.8) 26 (10.8) 50 (20.8) 5 (2.1) 16 (6.7) 7 (2.9) 18 (7.5) 11 (4.6)
On any immunosuppressive treatment *, *n* (%)	183 (76.3)
DNR orders, *n* (%)	114 (47.5)

All values expressed as mean (s.d.), unless otherwise specified. BMI: body mass index; CHF: chronic heart failure; COPD: chronic obstructive pulmonary disease; ECOG: Eastern Cooperative Oncology Group; HPB: hepatobiliary; ENT: ear–nose–throat; DNR: do not resuscitate. * Steroids, chemotherapy or other immunosuppressive treatment—e.g., rituximab, methotrexate, etc.—six months prior to admission.

**Table 2 jcm-11-03472-t002:** Admission and prognostic data.

Variable	
LOS (days)	23.7 (28.1)
LOS ImCU (days)	5.1 (4.5)
LOS previous to admittance to ImCU (days)	6.95 (15.3)
Location before ImCU admission, *n* (%)	
● Emergency room ICU Conventional Other	74 (30.8) 18 (7.5) 143 (59.6) 5 (2.1)
Urgent admission to ImCU, *n* (%)	210 (87.5).
Main cause of admission to ImCU, *n* (%)	
● Sepsis Acute respiratory failure (excluding sepsis) Post-surgical/Complex patient monitoring Cardiovascular Neurologic Other (hidroelectrolitic disturbance, bleeding, etc.)	77 (32.1) 93 (38.7) 22 (9.2) 19 (7.9) 10 (4.2) 19 (7.9)
Respiratory insufficiency, *n* (%) *	157 (65.4)
Use of non-invasive mechanical ventilation, *n* (%)	88 (36.7)
Use of vasoactive drugs, *n* (%)	26 (10.8)
Reason for discharge from ImCU, *n* (%)	
● Improvement Worsening Death Other	168 (70) 36 (15) 26 (10.8) 10 (4.2)
Hemoglobin (g/dL) **	10.05 (1.8)
Leucocytes (10 × 10^9^/L) **	11.3 (8.6)
Platelet count (10 × 10^9^/L) **	222 (163.6)
Reactive C protein (mg/dL) (NV < 0.4) **	15.4 (11.5)
Procalcitonin (ng/mL) ** (*n* = 190)	7.8 (20.5)
Creatinine (mg/dL) **	1.26 (1)
Urea (mg/dL) **	0.57 (0.40)
Albumin (g/dL) ** (*n* = 137)	2.35 (0.6)
Bilirubin (mg/dL) ** (*n* = 224)	2.2 (3.9)
SAPS 3 score (probability of in-hospital death, %)	68 (51.9)
In-hospital mortality, *n* (%)	90 (37.5)
SMR	0.72
ImCU mortality, *n* (%)	26 (10.8)
30 days mortality, *n* (%)	125 (52.1)
Survival after discharge (median, days) ^$^	81 (30.75–391.25)

All values expressed as mean (s.d.), unless otherwise specified. * patients with SpO2 <90% regardless of the main cause of admission (e.g., pleural effusion, pneumonia, fluid overload, pulmonary embolism, etc.), ** at ImCU admission. ^$^ Eighteen patients lost in follow-up. Date of last visit was used as reference. LOS: length of stay; ImCU: intermediate care unit; ICU: intensive care unit; NV: normal value; SAPS: Simplified Acute Physiology Score; SMR: standardized mortality ratio.

**Table 3 jcm-11-03472-t003:** Comparison between patients with and without do not resuscitate orders.

	DNR Orders	No DNR Orders	*p*-Value
Patients, *n*	114	126	
Male, *n* (%)	79 (69.3)	80 (63.5)	0.342
Age	65.25 (12.1)	60 (13.6)	0.002
Diabetes Mellitus, *n* (%)	20 (17.5)	33 (26.2)	0.107
CHF and/or Ischemic heart disease, *n* (%)	14 (12.2)	24 (19)	0.174
COPD, *n* (%)	24 (21.1)	16 (12.7)	0.083
Hypertension, *n* (%)	43 (37.7)	42 (33.3)	0.083
Cirrhosis, *n* (%)	12 (10.5)	6 (4.8)	0.090
ECOG ≥ 3, *n* (%)	50 (43.5)	34 (27)	0.006
Charlson comorbidity index score ≥ 7, *n* (%)	104 (91.2)	97 (77)	0.003
Partially dependent/Dependent, *n* (%)	105 (92.1)	106 (84.1)	0.058
On any immunosuppressive treatment *	90 (78.9)	93 (73.8)	0.350
LOS (days)	21.1 (18.4)	26.2 (34.5)	0.150
LOS ImCU (days)	5.8 (4.7)	4.5 (4.3)	0.023
Main cause of admission to ImCU, *n* (%)			0.015
● Sepsis Acute respiratory failure (excluding sepsis) Post-surgical/Complex patient monitoring Cardiovascular Neurologic Other (hidroelectrolitic disturbance, bleeding …)	41 (36) 49 (43) 5 (4.4) 4 (3.5) 5 (4.4) 10 (8.7)	36 (28.6) 44 (34.9) 19 (15.1) 15 (11.9) 5 (3.9) 7 (5.6)	
Respiratory insufficiency, *n* (%) **	79 (69.3)	78 (61.9)	0.229
Need for non-invasive mechanical ventilation, *n* (%)	51 (44.7)	37 (29.4)	0.014
Use of vasoactive drugs, *n* (%)	20 (17.5)	6 (4.8)	0.001
Reason for discharge from ImCU, *n* (%)			<0.001
● Improvement Worsening Death Other	69 (60.5) 17 (14.9) 24 (21.1) 4 (3.5)	99 (78.6) 19 (15.1) 2 (1.6) 6 (4.7)	
ImCU mortality, *n* (%)	24 (21.1)	2 (1.6)	<0.001
SAPS 3 Score (probability of in-hospital death, %)	73 (59.8)	64 (44.8)	<0.001
In-hospital mortality, *n* (%)	60 (52.6)	30 (23.8)	<0.001
30-day mortality, *n* (%)	79 (69.3)	45 (36.5)	<0.001
Survival after discharge (median, days) ^$^	46 (19.5–92.25)	162 (39.5–632)	<0.001

All values expressed as mean (s.d.), unless otherwise specified. * Steroids, chemotherapy or other immunosuppressive treatment—e.g., rituximab, methotrexate, etc.—six months prior to admission; ** patients with SpO2 <90% regardless of the main cause of admission (e.g., pleural effusion, pneumonia, fluid overload, pulmonary embolism, etc.); ^$^ eighteen patients lost in follow-up (4 in DNR group and 14 in no DNR orders group). Date of last visit was used as reference. DNR: do not resuscitate; CHF: chronic heart failure; COPD: chronic obstructive pulmonary disease; ECOG: Eastern Cooperative Oncology Group; LOS: length of stay; ImCU: intermediate care unit; SAPS: Simplified Acute Physiology Score.

**Table 4 jcm-11-03472-t004:** Multivariate analysis of predictors of hospital mortality.

Variable	OR [95% CI]	*p*-Value
DNR orders	2.4 [1.3–4.3]	0.004
SAPS 3 Score	1.02 [1–1.04]	0.002
ECOG performance status	2.3 [1.3–4.1]	0.006

CI: Confidence Interval; DNR: do not resuscitate. OR: odds ratio; SAPS: Simplified Acute Physiology Score; ECOG: Eastern Cooperative Oncology Group (3–4 vs. 0–2).

**Table 5 jcm-11-03472-t005:** Multivariate analysis of predictors of 30-day mortality.

Variable	OR [95% CI]	*p*-Value
DNR orders	3.6 [2–6.4]	0.000
Need for non-invasive mechanical ventilation	2.2 [1.2–4.1]	0.008
Bilirubin (mg/dl)	1.1 [1–1.2]	0.015

CI: Confidence Interval; DNR: do not resuscitate. OR: odds ratio.

## Data Availability

The datasets generated and/or analyzed during the current study are not publicly available but are available from the corresponding author on reasonable request.

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
