# Peer review of "Long-Term Outcome of Critically Ill Advanced Cancer Patients Managed in an Intermediate Care Unit"

_jcm, 2022, doi:10.3390/jcm11123472_

Round 1
Reviewer 1 Report
Th authors describe the long term outcome of critically ill advanced cancer patients managed in an intermediate care unit, and this study is interesting raising a point about the need for this type of ICU .
In the part Introduction, is not clear what do you mean by nihilism , please clarify
In the part materiel and method ligne 85 the word "hospitalist " is unclear who is he ?
In the part result the table 3 compare patients with and without DNR order
but is not clear why this variable was chosen, as is associated with the end of live stage it seems . I will be interested to know if the studied population had access to palliative care...
In the part discussion there nothing about the age of the studied population really young
Reviewer 2 Report
Thank you for the opportunity to ready your manuscript. The research was very interesting.
Please consider the following suggestions:
1. pg. 3, line 65, "analyze the main patient characterstics..." patient needs to be singular.
2. same page, line 68 - modific should be the word modify
Materials and Methods
This section needs more information. All instruments are missing a brief description as well as validity and reliability scores. Please also provide references.
1. pg 3, line 96, you mention functional status but you do not state how you are measuring it, unless you are using the ECOG. The way it is written, it looks as though they are separate measurements. If they are not, the wording needs to be more clear.
2. Same page, line 97 - Charlson Score - needs to be Charlson Comorbidity Index Score.
Results
1. pg 4, line 118 - When starting a sentence, the number needs to be spelled out. Same for Line 120.
2. Table 1 - While it is obvious you are talking about patients, please put that in the title: General Patient Characteristics or, General Patient Demographics.
3. Also in Table 1 - Please title your columns like you did in Tables 3-5. Also, you do not need to put 'n (%)' in every row, if you put that at the top of the column. It's more clear to the reader.
4. pg 7, line 169 - please insert 'than those without a DNR' before (Table 3). You do state if very clearly in lines 178-9 of pg 8. Otherwise it is not a complete thought or comparison.
5. pg 8, line 187 - Delete 'Among the' and just say, "A subgroup of patients...." It did not make sense the way it was written.
6. pg 8, line 192 - "By univariate analysis" doesn't make sense. Please consider, "Univariate analyses showed an increase in in-hospital mortality for those patients with....." The same with line 198.
Discussion
1. pg 9, lines 230-235 - This explanation of the SAPS 3 needs to go in the methods section when describing the instruments. It doesn't belong in the discussion section.
References
Over 80% of the references are >5 years old, and 80% of those are >10 years old. There needs to be more updated references.
One other note - I suggest using fewer acronyms. There were so many that it was difficult to remember them all without having to go back and look them up.
